# Comparative Outcomes of Treatment Strategies for Traumatic Distal Humerus Physeal Separation in Children: A Systematic Review

**DOI:** 10.3390/jcm14062037

**Published:** 2025-03-17

**Authors:** Byron Chalidis, Dimitrios Rigkos, Sonia Giouleka, Charalampos Pitsilos

**Affiliations:** 11st Orthopaedic Department, Aristotle University of Thessaloniki, 57010 Thessaloniki, Greece; dimitrisrigos@hotmail.gr; 2Obstetrics and Gynaecology Department, Leeds Teaching Hospitals Trust, Leeds LS1 3EX, UK; soniagiouleka@outlook.com; 3Academic Department of Trauma and Orthopaedics, School of Medicine, University of Leeds, Leeds LS2 9LU, UK; chpitsilos@outlook.com

**Keywords:** distal humerus, physeal separation, physeal fracture, cubitus varus, carrying angle, abuse, nonaccidental trauma, birth trauma

## Abstract

**Background**: Distal humerus physeal separation (DHPS) presents a rare injury type in young children often associated with misdiagnosis and delayed treatment. The aim of this study was to summarize all the available current evidence regarding the management and outcome of DHPS and determine the incidence of complications and particularly the cubitus varus deformity and avascular necrosis of the trochlea. **Methods**: A systematic review was conducted under the PRISMA guidelines. Medline/Pubmed, Scopus, Web of Science, and Cochrane were searched for studies dealing with children suffering from traumatic DHPS. **Results**: Twelve studies with a total of 257 children with DHPS were included for analysis. The mean age was 16.8 months (range: 0.1–46 months) with a mean follow-up of 37 months (range: 0.5–516 months). Non-accidental trauma was reported in 17.2% and misdiagnosis at initial assessment in 7.8%. Closed reduction and percutaneous pinning (CRPP) was the treatment of choice in 54.4%, open reduction and percutaneous pinning (ORPP) in 26.5%, closed reduction and cast immobilization (CR+cast) in 10.9%, and cast immobilization without reduction in 8.2%. The average range of extension–flexion arc was 2.1° to 127.8° (range: −10–140°). The mean Bauman’s angle was 72.4° (range: 66–79°), the mean shaft–condylar angle was 43.8° (range: 25–59°), the mean humeral length was 21.9 cm (range: 15.5–25.8 cm), and the mean carrying angle was 5.1° (range: 16° varus–19° valgus). According to Flynn’s criteria, 85.2% of cases were classified as excellent or good. The ORPP technique was associated with excellent results, while the CR+cast treatment combination was correlated with the poorest outcome (*p* = 0.001). Cubitus varus occurred in 18.9% (34 cases) and was highly correlated with CR+cast or cast immobilization alone without fracture reduction (*p* = 0.014). Avascular necrosis of the trochlea was found in 3.9% (7 cases) and was mainly apparent after cast immobilization without reduction (*p* < 0.001). **Conclusions**: Post-traumatic cubitus varus deformity may be encountered in approximately one-fifth of young children with DHPS. Surgical intervention with either CRPP or ORPP is the most effective treatment approach, leading to superior functional outcome and a lower complication rate.

## 1. Introduction

Distal humerus physeal separation (DHPS), also known as transphyseal separation, is a rare fracture pattern observed in young children and was first described by Smith almost 200 years ago [1]. Traumatic epiphyseal injuries account for approximately 20% of all pediatric fractures, with the majority occurring in the upper limbs [2]. However, involvement of the elbow joint is uncommon, and the relevant incidence of DHPS is very low, representing only 0.7% of all epiphyseal fractures [3]. In newborns and infants, the fracture is a result of difficult delivery during cesarian section, traumatic vaginal childbirth, or child abuse. In contrast, accidental trauma is the most common mechanism of injury in toddlers [4,5,6,7,8].

The diagnosis of DHPS is quite challenging, as the secondary ossification centers of the distal humerus are still unossified in young children [9]. The ossification center of the capitellum appears first by the age of one year, with the radial head, medial epicondyle, trochlea, olecranon, and lateral epicondyle following [10]. Thus, the injury may be wrongly interpreted as elbow dislocation, supracondylar fracture, or soft tissue swelling, leading to significant delay in treatment and subsequent development of physeal arrest, alignment deformity, joint stiffness, and poor outcome [11]. On the other hand, the other common fractures of the elbow joint, supracondylar and radial head fractures, have a higher potential for remodeling and achieving a better functional outcome [12,13]. Standard elbow X-rays may not always provide a clear diagnosis, and therefore, additional imaging modalities, including ultrasonography, magnetic resonance imaging (MRI), and elbow arthrography, are frequently required to determine the location and extent of injury [14,15,16].

The available treatment options for DHPS include elbow immobilization in situ, closed reduction with cast immobilization (CR+cast), and closed (CRPP) or open reduction (ORPP) in combination with percutaneous pinning [17,18]. However, according to the current literature, there is no consensus on the optimal treatment method of displaced DHPS. The primary goal of fracture management is to accomplish and maintain acceptable alignment until fracture healing, which takes place over a period of two to three weeks [19]. Malreduction and malalignment may lead to cubitus varus deformity and avascular necrosis of the trochlea (ANT), with a rate ranging from 4% to 58% among studies [20,21]. However, to the best of our knowledge, neither the exact incidence nor the relationship with specific treatment options have been clearly defined.

The unique challenges regarding the best diagnostic and treatment approach of DHPS prompted us to perform a systematic review to summarize the current evidence regarding the management and outcome of injury, as well as to determine the prevalence of cubitus varus deformity and ANT.

## 2. Materials and Methods

### 2.1. Study Type

A systematic review was conducted under the Preferred Reporting Items for Systematic Review and Meta-Analysis (PRISMA) guidelines [22] and was registered in the International Prospective Register of Systematic Reviews (PROSPERO) with registration number CRD42025643095.

### 2.2. Search Strategy

Four electronic databases (Medline/Pubmed, Scopus, Web of Science, and Cochrane) were searched up to December 2024 to identify any original studies regarding the management of DHPS. The following keywords were applied: distal AND (humerus OR humeral) AND (epiphyseal OR physeal) AND (separation OR fracture).

### 2.3. Inclusion Criteria and Study Selection

All clinical trials concerning the outcome of operative and non-operative management of DHPS injuries were enrolled for further analysis. The inclusion criteria included clinical studies containing at least five patients younger than four years of age published in English-language journals after the year 2000. Studies regarding DHPS in the older-children population, case reports, review articles, and animal studies were excluded from further screening. The EndNote X9 software (Clarivate Analytics, Philadelphia, PA, USA) was used to remove duplicate studies from the four databases. Moreover, the references of the initial search were also independently reviewed for possible missing articles. All article titles and abstracts were screened for eligibility by two independent reviewers (C.P. and D.R.). Any disagreements or conflicts concerning the inclusion or exclusion of the studies were resolved through discussion and adjudication by the third reviewer (B.C.).

### 2.4. Data Extraction

Two authors (C.P. and D.R.) extracted the following data and information from each eligible article using a Microsoft Excel database: first author’s last name, year of publication, number of patients, age, gender, follow-up period, injury classification, treatment, outcome of treatment, patients with cubitus varus, and other complications.

The most commonly used classification systems were described by Salter-Harris (S-H) and DeLee. In the Salter-Harris classification system, the physeal fractures are divided in five types based on the involvement of the physis, metaphysis, and epiphysis [23]. Type I injury pattern includes fractures through the physis. In type II injury, a physeal shear is combined with a largely vertical metaphyseal bony fracture. In type III, a partial physeal shear and an epiphyseal fracture are apparent. In type IV, the fracture involves the physeal, metaphyseal, and epiphyseal regions. Finally, in type V injury, there is a physeal and metaphyseal crush injury. According to DeLee classification, the fractures are divided in three groups: group A includes S-H type I fractures in children younger than 1 year of age, with a non-ossified lateral condyle; group B includes S-H types I or II fractures with a small metaphyseal fragment in children 1 to 3 years old, with an ossified lateral condyle; and group C refers to S-H type II fractures with a large metaphyseal fragment [24,25]. Cubitus varus was defined as varus alignment of the elbow or as a decreased carrying angle of more than 15° compared to the unaffected side [20,26].

The primary outcome of the study focused on estimating the overall incidence of cubitus varus deformity after DHPS. Secondary outcomes encompassed the incidence of other complications such as ANT as well as the assessment of any correlation between different treatment strategies and clinical outcome or complication rate.

### 2.5. Assessment of Risk of Bias

The quality of the included studies was evaluated using the “NIH Quality Assessment Tool for Observational Cohort and Cross-Sectional Studies” and the “NIH Quality Assessment Tool for Case Series Studies” [27]. These tools allow the rater to assign a three-level quality score—“good”, “fair”, or “poor”—based on the consideration of 14 and 9 items, respectively. Good quality indicates “low” risk of bias, fair quality “moderate” risk of bias, and poor quality “significant” risk of bias.

### 2.6. Data Synthesis and Analysis

Once the data extraction was completed, the collected data were transcribed into SPSS (IBM Corp., Armonk, NY, USA, released 2017; IBM SPSS Statistics for Windows, version 25.0. IBM Corp., Armonk, NY, USA) and further analyzed. The frequencies of categorical variables were compared using the chi-square (*χ*^2^) test. Statistical significance was assumed at a *p*-value of less than 0.05.

## 3. Results

### 3.1. Search Results

The initial literature search yielded a total number of 654 articles. After removal of duplicates, 468 articles were selected for title and abstract screening. According to the inclusion and exclusion criteria, 396 studies were deemed ineligible for analysis, leaving a total of 72 studies. Of them, 47 case reports and 13 review articles were excluded. During the review process, one more article was found to be relevant, and it was included in the analysis. Finally, 13 studies [20,24,26,28,29,30,31,32,33,34,35,36,37] were incorporated into the review, as summarized in the PRISMA flowchart (Figure 1).

### 3.2. Risk of Bias

Five studies were assessed to have low risk of bias, while the rest of them had moderate risk. The level of evidence, the methodological quality, and the risk of bias of the included studies are described in Table 1.

### 3.3. Demographic Data

A total number of 266 children with DHPS were identified in the 13 selected studies. The mean age was 16.2 months (range: 0.1–46 months). Gender distribution was reported in 10 studies and 171 patients [20,24,26,28,29,30,31,34,36,37]. Of these, 105 (61.4%) were males and 66 (38.6%) were females. The side of DHPS was described in 7 studies involving 143 children [24,30,31,33,34,36,37]. The right elbow was injured in 86 (60.1%) patients and the left one in the remaining 57 (39.9%). The mechanism of injury was mentioned in 9 studies and 151 children [20,24,30,31,32,33,34,35,36]. The most common cause of fracture was accidental trauma (89 cases, 55.6%) [20,24,32,33,35,36], followed by birth trauma (45 cases, 28.1%) [30,31,32,33,34,35,36]. Non-accidental injury was also reported in 26 cases (16.3%) [32,33,36]. The demographic data of the included patients are summarized in Table 2.

### 3.4. Diagnosis and Classification

The imaging tests used for the diagnosis of the injury were described in 11 studies and 184 patients [24,28,29,30,31,32,33,34,35,36,37]. Routine elbow X-rays were performed in all but 5 cases (97.3%) [35], elbow joint ultrasound in 35 cases (19%) [30,33,34,35,36], magnetic resonance imaging (MRI) in 12 cases (6.5%) [30,33,34,35], and diagnostic arthrogram in 1 case (0.5%) [34]. In 20 out of the total 266 children, the initial diagnosis was incorrect (7.5%) [31,32,34,36,37]. In 17 of the 20 misdiagnosed fractures [31,32,34,36], the initial diagnosis was another fracture around the elbow joint (7 cases, 41.2%) [32,34,36], elbow dislocation (5 cases, 29.4%) [31,34,36], elbow pain without any fracture (3 cases, 17.6%) [34,36], and Erb’s palsy [34] or septic elbow arthritis [32] (one case each, 5.9% each). The follow-up period was reported in 12 studies, including 264 children, and had a mean value of 38.4 months (range: 0.5–516 months) [20,24,26,28,29,30,31,33,34,35,36,37]. It is worthwhile that in three of these studies [25,29,31], the follow-up was relatively short, at less than 2 years.

The DeLee classification system (original or modified) was mainly used for evaluation of the fracture pattern (4 studies, 109 patients) [24,26,28,37]. Twenty-five fractures (22.9%) were classified as type A, 31 (28.5%) as type B, and 53 (48.6%) as type C. The Salter-Harris classification system was used in 3 studies, including 40 children [20,24,28]. There were 6 type I fractures (15%) and 34 (85%) type II fractures. Gartlant classification [38] was also applied in one study [35], which recorded two type I, one type II, two type III, and four type IV fractures. Details regarding the utilized radiological diagnostic methods and the classification of DHPS injuries are provided in Table 3.

### 3.5. Treatment and Outcome

Among a total number of 266 injured children, the most commonly applied treatment method was CRPP (140 patients, 52.6%) [20,24,28,29,30,31,32,33,37]. Open reduction and percutaneous pinning (ORPP) (71 cases, 26.7%) [24,26,33,35,36], closed reduction and cast immobilization (CR+cast) (29 cases, 10.9%) [20,26,30,31,34,35], and cast immobilization without reduction (26 cases, 9.8%) were also performed [20,32,34,35,36].

The range of motion at the last follow-up was reported in 3 studies and 39 patients [24,28,37]. The average range of elbow extension–flexion arc was 2.1° (range: −10–22°) to 127.8° (range: 74–140°). The mean Bauman’s angle (2 studies, 22 patients) [28,30] and the mean shaft–condylar angle (3 studies, 38 patients) [24,28,30] were 72.4° (range: 66–79°) and 43.8° (range: 25–59°), respectively. The final humeral length was measured in two studies (28 patients) [24,28] and the relevant mean value was 21.9 cm (range: 15.5–25.8 cm). According to data from four studies (51 patients), the average carrying angle was calculated to be 5.1° (range: 16° varus–19° valgus) [20,24,28,37].

Regarding the patient-reported outcome measures (PROMs), the Mayo Elbow Performance Score (MEPS) was used in two studies (28 patients) [24,28]. The mean MEPS was found to be 88.7 out of 100 (range: 70–95). In two other studies, including 81 patients [26,37], the Flynn criteria were used to assess the limb function. The outcome was classified as excellent in 52 (64.2%) children, good in 17 (21%) children, fair in 2 (2.5%) children, and poor in 10 (12.3%) children. All patients were treated with CR+cast or CRPP or ORPP. Both CRPP and OPRR treatment options were correlated with an excellent outcome (*p* = 0.023 and *p* = 0.001, respectively), while a poor outcome was mainly observed after CR+cast (*p* = 0.001). No difference was detected between CRPP and ORPP methods (*p* = 0.685). The selected treatment methods for the treatment of DHPS and the final outcomes are summarized in Table 4.

### 3.6. Complications

The most frequent complications of DHPS were elbow cubitus varus deformity and ANT (Table 5).

The occurrence of the above complications was discussed in 11 studies (180 patients) [20,24,26,28,29,30,31,34,35,36,37], and the calculated rate was 18.9% for cubitus varus (34 cases) [20,24,26,28,29,31,34,35,36,37]) and 3.9% for ANT (7 cases) [20,26]). In 10 of these studies (164 patients) [20,24,26,28,29,30,31,34,35,37], the treatment method of DHPS was also specified. Cubitus varus was reported in 37.9% (11 out of 29 cases) after CR+cast [20,26,31,34,35], in 21.7% (13 out of 60 cases) after CRPP [20,24,28,37], in 18.2% (2 out of 11 cases) after cast immobilization without reduction [20], and in 9.4% (6 out of 64 cases) after ORPP [24,26]. Comparative analysis revealed that CR+cast, CRPP, and cast immobilization without reduction were associated with increased incidence of post-traumatic cubitus varus deformity (*p* = 0.014). The respective rate of ANT was found to be 18.2% (2 out of 11 cases) after cast immobilization without reduction [20], 10.3% (3 out of 29 cases) after CR+cast [20], 1.7% (1 out of 60 cases) after CRPP [20], and 1.6% (1 out of 64 cases) after ORPP [26]. Thus, cast immobilization without reduction was correlated with higher incidence of ANT compared to the other treatment options (*p* < 0.001). Moreover, no difference between CRPP and ORPP methods was observed (*p* = 0.861).

## 4. Discussion

According to the current systematic review, the treatment of DHPS in young children results in satisfactory functional outcomes. The worst results are observed after conservative treatment with cast immobilization even after closed reduction of the fracture. Non-operative treatment is also associated with higher complication rates and particularly with cubitus varus deformity. Notably, the overall incidence of cubital varus after DHPS is relatively high, with a mean value of 19.3%.

Distal humerus physeal separation is usually the result of accidental trauma, such as a fall from height or a motor vehicle accident, but child abuse should always be ruled out [33,39]. In comparison to supracondylar elbow fracture, DHPS is more frequently associated with nonaccidental trauma, particularly when concomitant injuries of the appendicular and axial skeleton may present, such as proximal humerus fractures, bucket-handle, and condylar fracture [33,36]. In this study, we found that DHPS may have occurred after nonaccidental elbow injury in 17.2%. Birth trauma during vaginal delivery or cesarean section presents another common mechanism of injury [40]. Although arm protrusion and twin gestation are considered predisposing risk factors for development of DHPS, the majority of cases have been described after uneventful deliveries [34,41].

On plain radiographs, indirect signs of elbow intraarticular fractures such as joint effusion and soft tissue swelling may not always be apparent [36] (Figure 2).

Although the physeal part of DHPS may remain undisplaced in relation to the humerus, a posteromedial or medial translation of the ulna is commonly observed [32]. Interestingly, anterior displacement of the epiphysis has also been reported [42]. In newborns and infants with unossified epiphysis, ultrasonography can be a valuable diagnostic tool, especially if plain radiographs are negative or inconclusive [43]. Magnetic resonance imaging (MRI) not only confirms the diagnosis in challenging or unclear cases, particularly when the distal region of the humerus has not been still ossified, but also shows the presence and extend of concurrent injuries [30]. In the present study, we found that ultrasound and MRI examinations were utilized in 16% and 4% of cases, respectively, due to the inherent difficulty in establishing the diagnosis. We believe that ultrasonography and MRI screening should be routinely integrated in the diagnostic protocol when a high suspicion of distal humerus injury and normal elbow radiographs is apparent.

Delayed management of DHPS is not rare and is highly related to misdiagnosis or parental negligence [37]. Elbow dislocation and supracondylar or epicondylar fractures are the most common initial incorrect diagnosis [32,36]. Our analysis revealed that 7.8% of cases were misdiagnosed at the initial assessment. In the older published studies, delayed diagnosed injuries were treated with cast immobilization, without any attempt at reduction even in cases of displacement [20]. Recent studies have shown that favorable outcomes in displaced fracture patterns can be achieved only with operative treatment, even when executed up to 20 days following injury [37]. After that period of time, significant bone callus formation impedes any attempt for successful reduction, and corrective osteotomy may be considered in the future for correction of secondary cubitus varus deformity.

In infants and toddlers, elbow arthrogram is usually performed during CRPP of DHPS to improve anatomic reduction in the distal humerus (Figure 3) [28].

However, Chou et al. [29] suggested that, instead of intraoperative arthrogram, fracture reduction could be safely evaluated by checking if the ulnar axis was within the boundaries of the medial and lateral humeral lines. The authors reported that the method was associated with decreased incidence of cubital varus deformity.

Treatment of DHPS should be personalized according to patient age and mechanism of injury [44]. Neonates with DHPS are generally treated with CR+cast [31,45,46,47]. In case of loss of reduction, new CR+cast or CRPP are generally recommended [31]. de Jager and Hoffman [48], in their case series of 12 patients with fracture separation of the distal humeral epiphysis, recommended CRPP for children under the age of two and closed reduction and immobilization for older patients. In contrast, McIntyre et al. [49] advocated that persistent displacement in young children can be effectively managed through bone remodeling and, therefore, a favorable outcome should be anticipated. Jacobsen et al. [34] found that infants with DHPS due to birth trauma can be successfully treated with cast immobilization for two to four weeks, keeping the elbow at 90° of flexion and the forearm in pronation. No additional manipulation or other intervention were suggested. According to the current review and the included studies, closed reduction and percutaneous pinning is the most popular treatment option among physicians, as it has been selected in more than half of the young patients.

The prognosis of the treatment of DHPS could be considered satisfactory. Except for the radiological parameters, functional outcome measures have also been used for the evaluation of treatment effectiveness [50]. Poor outcomes are greatly associated with decreased carrying angle and cubitus varus deformity [29,34]. The incidence of cubitus varus was found to be almost 20% and was mainly observed after conservative treatment, including cast immobilization with or without fracture reduction. In a retrospective study, Chen et al. [26] compared the outcomes of 70 patients with DHPS treated with either CR+cast or ORPP. The authors found lower incidence of cubital varus deformity and improved function, based on Flynn’s criteria, after open reduction through a mini anterior transverse approach and K-wire fixation compared to conservative treatment. Elbow malalignment following conservative or operative treatment of DHPS has also been reported in some case reports [9,45,51,52]. However, these studies were not considered eligible for further analysis due to their short follow-up period of less than a year.

Regarding other risk factors for unsatisfactory results, Wu et al. [37] found that increased age of children and therapeutic delay were associated with limited range of motion, decreased carrying angle, and a worse Flynn’s rating after CRPP. On the other hand, Zhou et al. [24] found no correlation between patients’ age or time interval from trauma to surgery and MEPS score. Additionally, the authors reported that the functional outcome was graded as good regardless the direction of displacement, fracture pattern, and type of surgery—closed or open. Tudisco et al. [53] studied the long-term outcomes of five school-age children suffering from DHPS. After a mean follow-up of 45 years, the functional and cosmetic results were highly correlated with the child’s age at the time of fracture, the magnitude of initial displacement, and the quality of reduction. Although four out of five patients had a poor or fair result and joint degeneration was apparent after late or inadequate fracture reduction, no functional disability was reported even after heavy manual activity.

Avascular necrosis of the trochlea is one of the most devastating complications of DHPS, resulting in cubitus varus deformity [20]. According to the findings of the current study, ANT can develop in 4.1% of patients with DHPS. Residual varus deformity may necessitate operative correction with supracondylar lateral closing wedge osteotomy or dome and step-cut osteotomies [54]. Abe et al. [55] reported nine cases with significant cubitus varus angulation secondary to DHPS who were treated with supracondylar closing wedge osteotomies. Although the carrying angle was improved in all cases, one patient developed persistent varus deformity. However, all nine patients fully regained range of motion in the elbow.

This systematic review is subject to inherent limitations. Only studies with level IV evidence were integrated in the data analysis process, as no randomized control trials or comparative studies were identified during the database search. Therefore, it is not feasible to define a standardized treatment protocol across different healthcare settings and demographic groups. Furthermore, and due to limited information and lack of homogeneous data, it was difficult to assess the correlation between the values of carrying angle or cubitus varus deformity and the functional outcome. For the same reason, a subgroup analysis of different types of DHPS could not be carried out. However, the provided data from the included 257 young patients with DHPS were found adequate to determine the incidence of cubitus varus deformity and identify the most effective treatment strategy of DHPS.

## 5. Conclusions

Surgical intervention with either CRPP or ORPP is the most effective treatment approach for DHPS, leading to superior functional outcome and a lower complication rate. Post-traumatic cubitus varus deformity is not a rare event, as it may be encountered in approximately one-fifth of cases. The small number of patients in the published studies, the different classification systems, and the limited data regarding the correlation between specific fracture types and treatment outcomes do not allow for deeper analysis of fracture characteristics or the introduction of a specific algorithm for management. Further research with randomized control trials is required to enable the extraction of more robust conclusions regarding the optimal diagnostic approach and treatment of distal humerus physeal separation injuries.

## Figures and Tables

**Figure 1 jcm-14-02037-f001:**
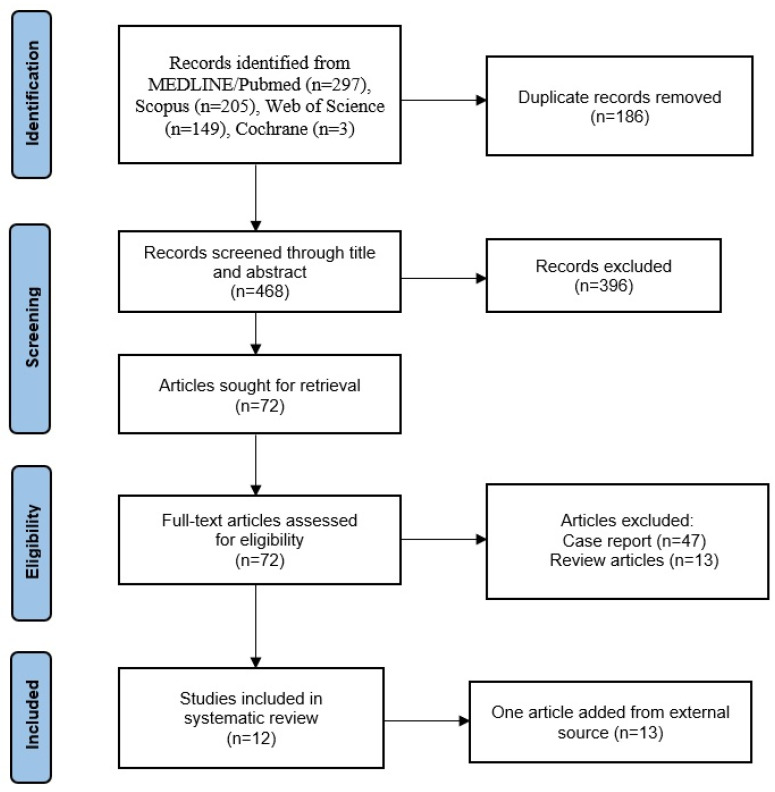
PRISMA flow diagram with research results. Thirteen eligible articles were included in the systematic review.

**Figure 2 jcm-14-02037-f002:**
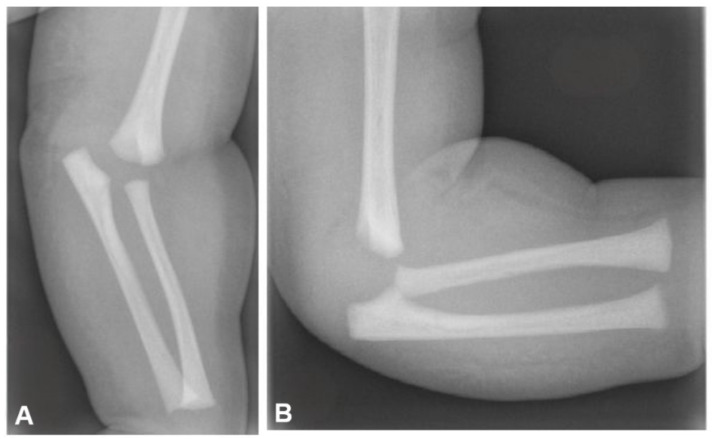
Plain elbow X-rays showing distal humerus physeal separation in an infant. Note that secondary ossification centers have not been ossified. (**A**) Anteroposterior view, (**B**) lateral view.

**Figure 3 jcm-14-02037-f003:**
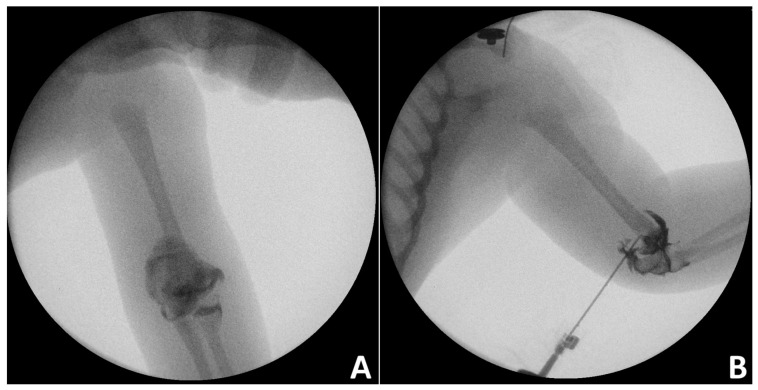
Intraoperative elbow arthrography images of an infant with distal humerus physeal separation (**A**) Anteroposterior view, (**B**) lateral view.

**Table 1 jcm-14-02037-t001:** Level of evidence, quality, and risk of bias assessment of the included studies.

Author	Year	Type of Study	Level of Evidence	Quality of Study	Risk of Bias
Cha [28]	2019	Case series	IV	Good	Low
Chen [26]	2022	Retrospective cohort study	IV	Good	Low
Chou [29]	2018	Case series	IV	Good	Low
Galeotti [30]	2023	Case series	IV	Fair	Moderate
Gigante [31]	2017	Case series	IV	Fair	Moderate
Gilbert [32]	2007	Case series	IV	Fair	Moderate
Hariharan [33]	2019	Case series	IV	Fair	Moderate
Jacobsen [34]	2009	Case series	IV	Fair	Moderate
Kruse [35]	2024	Case series	IV	Fair	Moderate
Oh [20]	2000	Case series	IV	Fair	Moderate
Supakul [36]	2015	Case series	IV	Fair	Moderate
Wu [37]	2024	Case series	IV	Good	Low
Zhou [24]	2019	Case series	IV	Good	Low

**Table 2 jcm-14-02037-t002:** Demographic data of the included studies.

Author	Year	Number of Patients	Mean Age (Months) (Range)	Gender (Males/ Females)	Side (Right/Left)	Mechanism	Follow-Up (Months) (Range)
Cha [28]	2019	12	23.5	4/8			116.2 (84–152)
Chen [26]	2022	70	19.6	46/24			44 (12–70)
Chou [29]	2018	13	20.4 (7–36)	5/8			13.1 (1–74)
Galeotti [30]	2023	10	0.1 (0–0.3)	7/3	5/5	10 birth trauma	37 (12–120)
Gigante [31]	2017	5	0.1 (0–0.1)	5/0	4/1	5 birth trauma	30 (12–60)
Gilbert [32]	2007	7	14.8 (0.5–23)			5 accidental trauma 1 birth trauma 1 abuse	
Hariharan [33]	2019	79	17.9 (0–46)		50/29	49 accidental trauma 21 nonaccidental trauma 9 birth trauma	2 (0.5–83.4)
Jacobsen [34]	2009	6	0.4 (0–1)	2/4	4/2	6 birth trauma	58 (16–120)
Kruse [35]	2024	9	0.6 (0–3)			6 birth trauma 1 accidental trauma	79 (24–201)
Oh [20]	2000	12	18.9 (13–36)	8/4		12 accidental trauma	23.5 (12–46)
Supakul [36]	2015	16	8.6 (0–28)	10/6	11/5	6 accidental trauma 6 birth trauma 4 abuse	5.6 (0.5–14)
Wu [37]	2024	11	14.3 (0–20)	8/3	5/6		43.3 (10–83)
Zhou [24]	2019	16	18 (11–37)	10/6	7/9	16 accidental trauma	42.3 (6–98)

**Table 3 jcm-14-02037-t003:** Imaging studies used for the diagnosis of distal humerus physeal separation and fracture classification.

Author	Number of Patients	Diagnosis	Number of Misdiagnoses	Classification (NoP per Type)
Cha [28]	12	12 X-rays	0	Mod DeLee (2 A, 7 B, 3 C) S-H (3 I, 9 II)
Chen [26]	70		0	DeLee (16 A, 14 B, 40 C)
Chou [29]	13	13 X-rays	0	
Galeotti [30]	10	10 X-rays 9 U/S 2 MRI	0	
Gigante [31]	5	5 X-rays	1 (elbow dislocation)	
Gilbert [32]	7	7 X-rays	4 (3 other fracture, 1 septic elbow arthritis)	
Hariharan [33]	79	79 X-rays 4 U/S 4 MRI	0	
Jacobsen [34]	6	6 X-rays 3 U/S 1 MRI 1 arthrogram	3 (1 elbow dislocation, 1 elbow pain, 1 Erb’s palsy)	
Kruse [35]	9	4 X-rays 7 U/S 5 MRI	0	Gartland (2 I, 1 II, 2 III, 4 IV)
Oh [20]	12		0	S-H (12 II)
Supakul [36]	16	16 X-rays 12 U/S	9 (4 other fracture, 3 elbow dislocation, 2 elbow pain)	
Wu [37]	11	11 X-rays	3 (not specified)	Mod DeLee (1 A, 7 B, 3 C)
Zhou [24]	16	16 X-rays	0	Mod DeLee (6 A, 3 B, 7 C) S-H (3 I, 13 II)

Abbreviations: MRI: magnetic resonance imaging, Mod: modified, NoP: number of patients, S-H: Salter-Harris classification, U/S: ultrasound.

**Table 4 jcm-14-02037-t004:** Treatment of distal humerus physeal separation and functional outcome.

Author	Number of Patients	Treatment (NoP)	Range of Motion	Radiographic Findings	Functional Outcome	Outcome
Cha [28]	12	12 CRPP	Mean f-e arch: 0.42–127.9°	Bauman’s angle: 71.7° Shaft–condylar angle: 43.3° Humeral length: 24 cm Carrying angle: 10.8°	MEPS: 92.9	Decreased carrying angle *. No difference * in f-e arch, Bauman’s angle, shaft–condyle angle, humeral length, MEPS.
Chen [26]	70	56 ORPP 14 CR+cast			Flynn’s criteria: 45 excellent, 13 good, 2 fair, 10 poor	Better functional outcome after ORPP. No difference in complication rate between two treatment methods.
Chou [29]	13	13 CRPP				No difference in ROM *
Galeotti [30]	10	6 CRPP 4 CR+cast		Bauman’s angle: 73.3° (range: 66–79) Shaft–condylar angle: 39.3° (range: 37–43)	Pain score: 6 score 0, 4 score 1	1 patient with reduced ROM * after CRPP
Gigante [31]	5	4 CR+cast 1 CRPP				No difference in ROM *.
Gilbert [32]	7	4 cast 3 CRPP				2 patients with extension loss (1 after cast, 1 after CRPP)
Hariharan [33]	79	77 CRPP 2 ORPP				4 with decreased ROM *
Jacobsen [34]	6	4 cast 2 CR+cast				1 with decreased ROM * after cast
Kruse [35]	9	5 cast 3 ORPP 1 CR+cast				No difference in ROM *.
Oh [20]	12	6 CRPP 4 CR+cast 2 cast		Carrying angle: 6.1° varus (range: 0–16 varus)		
Supakul [36]	16	11 cast 5 ORPP				No difference in ROM *
Wu [37]	11	11 CRPP	Mean f-e arch: −1.7–132.7°	Carrying angle: 5.7° (range: 0–12)	Flynn’s criteria: 7 excellent, 4 good	Increased age and injury to surgery interval are correlated with changes * in ROM, carrying angle and Flynn’s rating.
Zhou [24]	16	11 CRPP 5 ORPP	Mean f-e arch: 5.9–124.4°	Shaft–condylar angle: 47.1° (range: 25–59) Humeral length: 20.3 cm (range: 15.5–25.8) Carrying angle: 8.8° (range: 2–19)	MEPS: 85.5 (range; 70–95)	Difference * in carrying angle and MEPS score. No difference * in humeral shaft–condylar angle. 15/16 returned to everyday activity and sport.

Abbreviations: CR: closed reduction, CR+cast: closed reduction and cast immobilization, CRPP: closed reduction and percutaneous pinning, f-e: flexion–extension, MEPS: Mayo Elbow Performance Score, NoP: number of patients, ORPP: open reduction and percutaneous pinning, ROM: range of motion. * compared to unaffected side.

**Table 5 jcm-14-02037-t005:** Complications of distal humerus physeal separation.

Author	Number of Patients	Treatment (NoP)	Number of Cubitus Varus	Other Complications
Cha [28]	12	12 CRPP	2 (2 CRPP)	0
Chen [26]	70	56 ORPP 14 CR+cast	9 (4 ORPP, 5 CR+cast)	1 ANT (ORPP)
Chou [29]	13	13 CRPP	1	
Galeotti [30]	10	6 CRPP 4 CR+cast	0	0
Gigante [31]	5	4 CR+cast 1 CRPP	1 (CR+cast)	0
Gilbert [32]	7	4 cast 3 CRPP	n/a	n/a
Hariharan [33]	79	77 CRPP 2 ORPP		6 cubitus varus or valgus, 1 additional surgery (distal humeral osteotomy for varus deformity)
Jacobsen [34]	6	4 cast 2 CR+cast	1 (CR+cast)	0
Kruse [35]	9	5 cast 3 ORPP 1 CR+cast	1 (CR+cast)	
Oh [20]	12	6 CRPP 4 CR+cast 2 cast	7 (2 CRPP, 3 CR+cast, 2 cast)	6 ANT (1 CRPP, 3 CR+cast, 2 cast)
Supakul [36]	16	11 cast 5 ORPP	2	0
Wu [37]	11	11 CRPP	2 (2 CRPP)	0
Zhou [24]	16	11 CRPP 5 ORPP	8 (6 CRPP, 2 ORPP)	0

Abbreviations: ANT: avascular necrosis of trochlea, CR: closed reduction, CR+cast: closed reduction and cast immobilization, CRPP: closed reduction and percutaneous pinning, NoP: number of patients, ORPP: open reduction and percutaneous pinning, n/a: not applicable.

## Data Availability

No new data were created or analyzed in this study. Data sharing is not applicable to this article.

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
