# Peer review of "Comparative Outcomes of Treatment Strategies for Traumatic Distal Humerus Physeal Separation in Children: A Systematic Review"

_jcm, 2025, doi:10.3390/jcm14062037_

Round 1

Reviewer 1 Report

Comments and Suggestions for Authors

We thank the authors for this review entitled: "Traumatic Distal Humerus Physeal Separation".

The PRISMA methodology was well conducted, however I have some comments.

1- Line 51: I would add the age of the lateral condyl ossification, which is of interest regarding the articles selected for the review. 

2- The authors state  Line 54-55: "This contrasts with other common fractures around
the elbow, which have the potential to remodel and achieve a satisfactory functional recovery". Which are the fractures the authors are mentionning? Indeed, the remodeling potential of the elbow is 20%, regardless of the bone.

3- Line 63-65: "within 2 to 3 weeks" is it referring to the fracture healing or the acceptable alignment? Maybe rephrase the sentence to clarify.

4- Line 65-66: can the authors provide a range for each complication? Although cubitus varus is a common complication, I do not expect avascular necrosis in those fractures except secondary to iatrogenic surgical procedures with posterior approaches.

5- Line 266-267: the sentence must be rephrased in order to improve the syntax " In compared to the other common elbow injury pattern the supracondylar elbow 266
fracture, DHPS is more frequently associated with nonaccidental trauma particularly 267
when concomitant injuries of the appendicular and axial skeleton may present such as 268
proximal humerus fractures, bucket-handle and condylar fracture [8][29,31]."

6- US and MRI are of particular interest in DeLee stage A fracture, when the lateral condyl is not yet ossified. Similarly, "the delayed management" is more frequent in Stage A fractures, for the same reason. Furthermore, the discussion largely debate the management of Stage A fracture, even though a minority of the patients (23%) were reported with this stage. I would recommend to the authors to analyze Stage A fractures separately from Stage B and C since, the management and diagnosis is more difficult due to the absence of the lateral condyl ossification node.

7- The authors state Lines 116-118: "The primary outcome of the study focused on estimating the overall incidence of cubitus varus after DHPS. Secondary outcomes encompassed the incidence of other complications such as avascular necrosis of the trochlea and the assessment of any correlation between different treatment strategies and clinical outcome – complication rates. " However, this is not quite consistent with the aim The aim of this systematic review was to summarize the current evidence regarding the management and outcome of DHPS, as well as to determine the incidence of complications, including the cubitus varus and ANT" (Lines 69-71). The majority of the results and the discussion are also focused on the diagnosis and the management. Should the complications be considered as the primary outcomes? If the outcomes of this review are modified, the conclusion should also be modified accordingly.

8- I suggest the authors to consider to add the following article in their review: "Management of the supposed elbow dislocation in newborns" J Child Orthop. 2024 Dec 10;19(1):48-55.

Comments on the Quality of English Language

Some synthax and grammatic errors to improve

Author Response

Dear Editor,

We would like to thank you for accepting to reconsider our manuscript titled: “Comparative Outcomes of Treatment Strategies for Traumatic Distal Humerus Physeal Separation in Children: A Systematic Review” for publication in the “Journal of Clinical Medicine” journal.

We would also like to thank the reviewers for their insightful comments. All raised points have been addressed and the manuscript has been revised according to their suggestions. All text changes in the manuscript have been highlighted. For reviewing purposes, the comments have been addressed one by one. Be aware that as one reviewer suggested adding another study in the analysis, the result section has been changed.

Specifically:

Reviewer #1

Comment: 1- Line 51: I would add the age of the lateral condyle ossification, which is of interest regarding the articles selected for the review. 

Reply: According to reviewer suggestion,  a relevant comment and a citation have been added in the manuscript. (Lines: 55-57)

Comment: 2: The authors state  Line 54-55: "This contrasts with other common fractures around
the elbow, which have the potential to remodel and achieve a satisfactory functional recovery". Which are the fractures the authors are mentioning? Indeed, the remodeling potential of the elbow is 20%, regardless of the bone.

Reply: Thank you for the comment. We have provided information regarding specific fracture types around the elbow joint that have the potential to remodel. We have also added a citation (No 13) of a study that showed more than 90% chance of remodeling after malunion of pediatric supracondylar fractures, when patients are younger than 8 years of age. (Lines: 60-62)

Comment 3:  Line 63-65: "within 2 to 3 weeks" is it referring to the fracture healing or the acceptable alignment? Maybe rephrase the sentence to clarify.

Reply: Thank you for the comment. We have re-phrased the sentence according to your suggestion. (Lines: 70-72)

Comment 4:  Line 65-66: can the authors provide a range for each complication? Although cubitus varus is a common complication, I do not expect avascular necrosis in those fractures except secondary to iatrogenic surgical procedures with posterior approaches.

Reply:  Thank you for the comment. The range of the reported complications has been added in the text (Lines: 72-74)

Comment 5: Line 266-267: the sentence must be rephrased in order to improve the syntax " In compared to the other common elbow injury pattern the supracondylar elbow
fracture, DHPS is more frequently associated with nonaccidental trauma particularly when concomitant injuries of the appendicular and axial skeleton may present such as
proximal humerus fractures, bucket-handle and condylar fracture [8][29,31]."

Reply: Thank you for the comment. The relevant sentence has been re-written accordingly. (Lines: 282-285).

Comment 6: US and MRI are of particular interest in DeLee stage A fracture, when the lateral condyle is not yet ossified. Similarly, "the delayed management" is more frequent in Stage A fractures, for the same reason. Furthermore, the discussion largely debate the management of Stage A fracture, even though a minority of the patients (23%) were reported with this stage. I would recommend to the authors to analyze Stage A fractures separately from Stage B and C since, the management and diagnosis is more difficult due to the absence of the lateral condyle ossification node.

Reply: Thank you for the comment. The value of MRI has been further pointed out in young children with unossified epiphyses. Unfortunately, a subgroup analysis of different types of fractures according to DeLee classification could not be carried out due to the limited data from the included studies. A relevant comment has been added in the paragraph describing the limitations of the systematic review. (Lines: 387-388)

Comment 7: The authors state Lines 116-118: "The primary outcome of the study focused on estimating the overall incidence of cubitus varus after DHPS. Secondary outcomes encompassed the incidence of other complications such as avascular necrosis of the trochlea and the assessment of any correlation between different treatment strategies and clinical outcome – complication rates. " However, this is not quite consistent with the aim The aim of this systematic review was to summarize the current evidence regarding the management and outcome of DHPS, as well as to determine the incidence of complications, including the cubitus varus and ANT" (Lines 69-71). The majority of the results and the discussion are also focused on the diagnosis and the management. Should the complications be considered as the primary outcomes? If the outcomes of this review are modified, the conclusion should also be modified accordingly.

Reply: Thank you for the comment and the suggestions. The objective of the study is to present the current evidence regarding the management and outcomes of distal humerus physeal separation. For this reason, the diagnostic approaches and treatment options have been extensively analyzed. However, and in accordance with your comment, we had to define the primary and secondary outcomes of the study. Therefore, the incidence of cubitus varus deformity and avascular necrosis  is considered the primary aim of the study and the correlation between complications and the type of management the secondary outcome. According to your recommendation, we have changed the conclusion accordingly. (Lines: 394-401)

Comment 8: I suggest the authors to consider to add the following article in their review: "Management of the supposed elbow dislocation in newborns" J Child Orthop. 2024 Dec 10;19(1):48-55.

Reply: Thank you for the comment. This study has been added in the manuscript and the “Results” section has been modified accordingly.

Reviewer 2 Report

Comments and Suggestions for Authors

Thank you for the opportunity to review your manuscript entitled "Traumatic Distal Humerus Physeal Separation. A Systematic Review." This manuscript provides valuable insights into the treatment strategies for distal humerus physeal separation in children. Below are some suggestions aimed at further enhancing the clarity and depth of your manuscript:

General Comments:

  • The manuscript is well-structured and informative. However, incorporating discussions about the feasibility of treatment strategies across different healthcare settings could significantly enhance its practical applicability and relevance.
  • A deeper analysis of patient-reported outcomes in relation to specific treatment modalities would strengthen the clinical applicability and help clinicians better understand the impact of these treatments from the patient's perspective.

Specific Comments:

Title:

  • Consider refining the title to more precisely reflect the emphasis on specific treatment outcomes, e.g., "Comparative Outcomes of Treatment Strategies for Traumatic Distal Humerus Physeal Separation in Children: A Systematic Review."

Abstract:

  • The abstract should include key findings related to the efficacy of treatment methods and the prevalence of specific complications such as cubitus varus and avascular necrosis, providing a succinct yet comprehensive overview of the study's outcomes.

Introduction:

  • Lines 40-50: Expand the discussion on the clinical and diagnostic challenges associated with distal humerus physeal separation to better justify the need for this systematic review.

Methods:

  • Lines 73-91: Clearly articulate the inclusion criteria and explain the rationale behind the study selection process:
    • Explain the criteria for prioritizing certain types of studies and the quality assessment methods used for including studies in the review.

Results:

  • Lines 137-165: Provide detailed comparisons of outcomes associated with different treatment modalities such as closed reduction with percutaneous pinning versus open reduction, highlighting scenarios in which one might be preferred over the other based on clinical indications or patient specifics.

Discussion:

  • Lines 256-280: Discuss the limitations in the context of applying findings universally across different populations and healthcare systems.
  • Add references to recent systematic reviews or meta-analyses that either support or contrast with your findings to provide a richer context.
  • Lines 318-335: Elaborate on the long-term effects of the treatment modalities discussed, with a focus on quality of life and functional outcomes for patients.

Tables and Figures:

  • Table 2: Include a new column summarizing the specific surgical interventions used and their outcomes to enhance clarity and quick reference for readers.
  • Figure 1: Improve the caption to more comprehensively explain the statistical methods used in the flow diagram and the relevance of the presented data.

Conclusion:

  • Emphasize the importance of timely and appropriate intervention in improving outcomes and reducing complications for pediatric patients with distal humerus physeal separations.
  • Suggest prospective areas for further research, such as the impact of standardized treatment protocols and the role of emerging imaging technologies in treatment planning.

Reference Enhancements:

  1. Comparative Therapeutic Approaches:
    • Include "Osteoarthritis: a call for research on central pain mechanism and personalized prevention strategies" to emphasize the need for personalized treatment strategies in managing distal humerus physeal separations, drawing parallels with osteoarthritis care.
  2. Support for Methodological Choices:
    • Reference "Reliability of pinch strength testing in elderly subjects with unilateral thumb carpometacarpal osteoarthritis" to justify the choice of functional outcome measures in your study. This source supports the use of validated tests to assess treatment outcomes, applicable to evaluating pediatric orthopedic treatments.
Comments on the Quality of English Language

The manuscript is proficiently crafted, yet there is room for minor improvements to augment clarity and enhance the readability:

Grammar and Syntax:The manuscript would benefit from simplifying complex sentence structures to aid comprehension. Consistency in verb tenses across the document should also be ensured for coherence.
Technical Terminology:It is recommended to define all specialized terms upon their first appearance within the text. Additionally, including a glossary could provide valuable assistance to readers unfamiliar with specific jargon.
Punctuation and Style:A detailed review of punctuation is advised to correct any inaccuracies. Employing an active voice more frequently can make the narrative more engaging and direct.
Spelling and Typos:A comprehensive spell-check is necessary to fix any typographical errors. Consistency in the spelling of technical terminology will also reinforce professional integrity and clarity.
Overall Readability:Enhancing the transitions between sections and utilizing subheadings more effectively can significantly improve the manuscript's flow and make it easier for readers to navigate through the content.

Author Response

Dear Editor,

We would like to thank you for accepting to reconsider our manuscript titled: “Comparative Outcomes of Treatment Strategies for Traumatic Distal Humerus Physeal Separation in Children: A Systematic Review” for publication in the “Journal of Clinical Medicine” journal.

We would also like to thank the reviewers for their insightful comments. All raised points have been addressed and the manuscript has been revised according to their suggestions. All text changes in the manuscript have been highlighted. For reviewing purposes, the comments have been addressed one by one. Be aware that as one reviewer suggested adding another study in the analysis, the result section has been changed.

Specifically:

Reviewer #2

Comment: The manuscript is well-structured and informative. However, incorporating discussions about the feasibility of treatment strategies across different healthcare settings could significantly enhance its practical applicability and relevance.

Reply: We fully agree with this comment and therefore the paragraph regarding the “limitations of study” has been modified accordingly. Unfortunately, and due to lack of adequate data, a standardized general protocol could not be clearly defined. (Lines: 383-385)

Comment: A deeper analysis of patient-reported outcomes in relation to specific treatment modalities would strengthen the clinical applicability and help clinicians better understand the impact of these treatments from the patient's perspective.

Reply: Thank you for the comment. Tables 4 and 5 provide data regarding the relationship between the applied treatment approaches and the functional outcomes and complications. As mentioned above, the small number of patients in each study, the different classification systems and the limited data regarding the correlation between specific fracture types and treatment outcomes do not allow a deeper analysis and introduction of a specific algorithm of management. This statement has been added in “Conclusion” section of the manuscript. (Lines: 398-401)

Comment: Consider refining the title to more precisely reflect the emphasis on specific treatment outcomes, e.g., "Comparative Outcomes of Treatment Strategies for Traumatic Distal Humerus Physeal Separation in Children: A Systematic Review."

Reply: Thank you for the comment The title was refined accordingly. (Lines: 2-4)

Comment: The abstract should include key findings related to the efficacy of treatment methods and the prevalence of specific complications such as cubitus varus and avascular necrosis, providing a succinct yet comprehensive overview of the study's outcomes.

Reply: Thank you for the comment. The Abstract section has been modified in order to better clarify the efficacy of treatment methods. Moreover, information about the other main complication, the avascular necrosis of the trochlea, is also provided. (Lines: 17-18 and 32-37)

Comment: Lines 40-50: Expand the discussion on the clinical and diagnostic challenges associated with distal humerus physeal separation to better justify the need for this systematic review.

Reply: Thank you for the comment. The relevant part in “Introduction” section of the article has been expanded in order to present the need for the conduction of the systematic review. (Lines: 74-79)

Comment: Lines 73-91: Clearly articulate the inclusion criteria and explain the rationale behind the study selection process: Explain the criteria for prioritizing certain types of studies and the quality assessment methods used for including studies in the review. (number of p and homogeneity of treatment strategies)

Reply: Thank you for the comment. Following your instructions, the inclusion criteria and the assessment of the quality of included studies are presented in 2.3 and 2.5 parts of “Materials-Methods” section of the article (Lines: 94, 96, 132-138)

Comment: Lines 137-165: Provide detailed comparisons of outcomes associated with different treatment modalities such as closed reduction with percutaneous pinning versus open reduction, highlighting scenarios in which one might be preferred over the other based on clinical indications or patient specifics.

Reply: Thank you for the comment. Tables 4 and 5 provide comparative data among the available studies regarding the outcomes and complications of applied treatment options. At the end of 3.4 and 3.5 paragraphs of “Results” section and according to the statistical analysis, the most and least preferred treatment modality are also presented. Regarding the comparison between the two main surgical options, closed reduction with percutaneous pinning versus open reduction and pinning, no difference in terms of functional outcome and avascular necrosis was identified. Increased incidence of cubitus varus was observed after closed reduction and percutaneous pinning. However, and due to the heterogeneity of the information found in the included studies, no further analysis of treatment options could be made. (Lines: 232-235, 263-265, 267-270)

Comment:  Lines 256-280: Discuss the limitations in the context of applying findings universally across different populations and healthcare systems.

Reply: Thank you for your comment. According to your suggestion, relevant changes have been made in “Discussion”(last paragraph) and “Conclusion” sections of the manuscript. (Lines: 383- 385, 398-401)

Comment: Add references to recent systematic reviews or meta-analyses that either support or contrast with your findings to provide a richer context.

Reply: To the best of our knowledge, there are no similar systematic reviews or meta-analysis published in any of the electronic databases searched. This was the main reason to utilize the current study. As a result, there is no high evidence literature to either support or contrast our findings.

Comment: Lines 318-335: Elaborate on the long-term effects of the treatment modalities discussed, with a focus on quality of life and functional outcomes for patients.

Reply: Thank you for the comment. More details about long-term functional outcome after treatment of DHPS have been added in the text. (Lines: 368-371)

Comment: Table 2: Include a new column summarizing the specific surgical interventions used and their outcomes to enhance clarity and quick reference for readers. (table 3)

Reply: Thank you for the comment. This information is presented in Table 4.

Comment: Figure 1: Improve the caption to more comprehensively explain the statistical methods used in the flow diagram and the relevance of the presented data. (12 studies were selected)

Reply:  According to your suggestion, the caption of Figure 1 has been changed accordingly. As one more study has been added in the manuscript the total number of included studies is now 13. (Lines: 152-155)

Comment: Emphasize the importance of timely and appropriate intervention in improving outcomes and reducing complications for pediatric patients with distal humerus physeal separations.

Reply: Thank you for the comment. According to your suggestion, the importance of early diagnosis has been emphasized and more information in Discussion section has been added. (Lines: 316-319)

Comment: Suggest prospective areas for further research, such as the impact of standardized treatment protocols and the role of emerging imaging technologies in treatment planning.

Reply: Thank you for the comment. The importance of current imaging modalities (US and MRI) and the need for their integration in the diagnosis of the injury have been inserted in the text. Furthermore, the necessity of future research to establish an evidence-based treatment approach is also presented in manuscript. (Lines: 299-308)

Comment: Include "Osteoarthritis: a call for research on central pain mechanism and personalized prevention strategies" to emphasize the need for personalized treatment strategies in managing distal humerus physeal separations, drawing parallels with osteoarthritis care.

Reply: Thank you for the comment. The suggested citation along with a relevant comment have been added in the manuscript (Discussion section). (Lines: 332-333)

Comment: Reference "Reliability of pinch strength testing in elderly subjects with unilateral thumb carpometacarpal osteoarthritis" to justify the choice of functional outcome measures in your study. This source supports the use of validated tests to assess treatment outcomes, applicable to evaluating pediatric orthopedic treatments.

Reply: Thank you for the comment. The suggested reference and a relevant comment have been added in the text (Discussion section). (Lines: 346-348)

Comment: The manuscript would benefit from simplifying complex sentence structures to aid comprehension. Consistency in verb tenses across the document should also be ensured for coherence.

Reply: Thank you for the comment. The whole document has been scanned in order to improve its content and simplify any complex tasks and phrases.

Comment: It is recommended to define all specialized terms upon their first appearance within the text. Additionally, including a glossary could provide valuable assistance to readers unfamiliar with specific jargon.

Reply: Thank you for the comment. Specialized terms have been clarified during their first appearance in text and a glossary has been inserted at the end of manuscript. (Lines: 431-445)

Comment: A detailed review of punctuation is advised to correct any inaccuracies. Employing an active voice more frequently can make the narrative more engaging and direct.

Reply: Thank you for both comments. The text has been re-evaluated and corrected to address the above issues.

Comment: A comprehensive spell-check is necessary to fix any typographical errors. Consistency in the spelling of technical terminology will also reinforce professional integrity and clarity.

Reply: Thank you for these comments also. As mentioned before, the whole manuscript has been re-checked to correct any inconsistencies, mistakes, grammatical and typographical errors.

Comment: Enhancing the transitions between sections and utilizing subheadings more effectively can significantly improve the manuscript's flow and make it easier for readers to navigate through the content.

Reply: Thank you for the comment. We have re-checked the manuscript to improve the manuscript’s flow and readability. All the sections with their subheadings have been prepared and numbered according to Journal’s instructions.

Reviewer 3 Report

Comments and Suggestions for Authors

The authors present a systematic review with pooled data analysis on distal humerus physeal separation in children. I find this review well-conducted, highly useful, and of general interest to readers. I have only a few suggestions:

  1. Title change: I suggest renaming it "Traumatic Distal Humerus Physeal Separation: A Systematic Review and Meta-analysis."

  2. Case report analysis: It would be valuable to check the number of patients and complication rates in the excluded case reports. In rare conditions like physeal separation, understanding complication types and frequencies from case reports is relevant. If complication rates in case reports significantly exceed those in larger studies, this should be highlighted.

  3. Imaging inclusion: Consider adding an X-ray (and possibly an MRI, CT, or ultrasound) of a patient with physeal separation.

  4. Quality assessment:

    • The tool cited in line 124 should be referenced properly, removing the web address.
    • Specify the minimum follow-up duration required for the highest quality rating. Studies with <2 years of follow-up should be downgraded, as they may underestimate post-traumatic deformities.
  5. Complication rate analysis: A more robust statistical method is needed to compare surgical vs. non-surgical outcomes. A forest plot (showing confidence intervals and study heterogeneity), logistic regressions, or mixed models would better assess the impact of multiple variables on complications. Lack of proper statistical analysis may introduce bias, reducing the review’s reliability and quality.

  6. Publication bias assessment: A funnel plot should be included.

  7. Meta-regression: It would be useful to evaluate whether age at trauma and follow-up duration influence complication rates (especially cubitus varus). Age at trauma may also affect treatment choice (e.g., obstetric trauma is less likely to be treated surgically).

Author Response

Dear Editor,

We would like to thank you for accepting to reconsider our manuscript titled: “Comparative Outcomes of Treatment Strategies for Traumatic Distal Humerus Physeal Separation in Children: A Systematic Review” for publication in the “Journal of Clinical Medicine” journal.

We would also like to thank the reviewers for their insightful comments. All raised points have been addressed and the manuscript has been revised according to their suggestions. All text changes in the manuscript have been highlighted. For reviewing purposes, the comments have been addressed one by one. Be aware that as one reviewer suggested adding another study in the analysis, the result section has been changed.

Specifically:

Reviewer #3

Comment: Title change: I suggest renaming it "Traumatic Distal Humerus Physeal Separation: A Systematic Review and Meta-analysis."

Reply: Thank you for the comment. According to our knowledge, the data of the study do not support the execution of a Meta-Analysis. Moreover, another reviewer suggested to change also the Title to point out the comparative results of the analysis (Comparative Outcomes of Treatment Strategies for Traumatic Distal Humerus Physeal Separation in Children: A Systematic Review). If you  disagree with the content, please let us know to proceed to further changes.

Comment: Case report analysis: It would be valuable to check the number of patients and complication rates in the excluded case reports. In rare conditions like physeal separation, understanding complication types and frequencies from case reports is relevant. If complication rates in case reports significantly exceed those in larger studies, this should be highlighted.

Reply: Thank you for the comment. The published case reports included patients with divergent data and almost no complications were reported. In the few studies where elbow deformity were recorded, the follow-up period was up to one year, considering them non-eligible for inclusion. A relevant comment has been inserted in the text. (Lines 355-358)

Comment: Imaging inclusion: Consider adding an X-ray (and possibly an MRI, CT, or ultrasound) of a patient with physeal separation.

Reply: Thank you for the comment. A radiographic image of distal humerus physeal separation has been added. (Lines: 293-295)

Comment: The tool cited in line 124 should be referenced properly, removing the web address.

Reply: Thank you for the comment. The web address has been removed from the text and the tool was properly cited. (Line: 135)

Comment: Specify the minimum follow-up duration required for the highest quality rating. Studies with <2 years of follow-up should be downgraded, as they may underestimate post-traumatic deformities.

Reply: Thank you for the comment. Three studies [25, 29 and 31] reported a follow up less than 2 years. A relevant comment has been added in the text. (Lines: 194-195)

Comment: Complication rate analysis: A more robust statistical method is needed to compare surgical vs. non-surgical outcomes. A forest plot (showing confidence intervals and study heterogeneity), logistic regressions, or mixed models would better assess the impact of multiple variables on complications. Lack of proper statistical analysis may introduce bias, reducing the review’s reliability and quality.

Reply: Thank you for the comment. Unfortunately, and according to our data and knowledge we are unable to perform a complication rate analysis with a forest plot and a meta-analysis.

Comment: Publication bias assessment: A funnel plot should be included.

Reply: Thank you for the comment. As mentioned above, we are unable  to create a funnel plot for assessing the risk of bias. All but one studies were of level of evidence IV, and this fact complicates the process even more.

Comment: Meta-regression: It would be useful to evaluate whether age at trauma and follow-up duration influence complication rates (especially cubitus varus). Age at trauma may also affect treatment choice (e.g., obstetric trauma is less likely to be treated surgically).

Reply: Thank you for the comment. Unfortunately, we could not perform further analysis and the inclusion of mainly level of evidence IV studies obscures the ability to perform the suggested analysis.

Round 2

Reviewer 1 Report

Comments and Suggestions for Authors

The article can be accepted in the current form. The authors have made the required improvements.